# The FDA-Approved Drug Pyrvinium Selectively Targets ER^+^ Breast Cancer Cells with High INPP4B Expression

**DOI:** 10.3390/cancers15010135

**Published:** 2022-12-26

**Authors:** Samuel J. Rodgers, Lisa M. Ooms, Christina A. Mitchell

**Affiliations:** Department of Biochemistry and Molecular Biology, Biomedicine Discovery Institute, Monash University, Clayton, VIC 3800, Australia

**Keywords:** breast cancer, ER, INPP4B, pyrvinium, tamoxifen

## Abstract

**Simple Summary:**

Wnt/β-catenin signaling is hyperactivated in many human cancers including up to 50% of breast cancers. Although there has been significant progress in developing therapeutics that suppress Wnt/β-catenin signaling, particularly in colon cancer, the repurposing of FDA-approved therapeutics is likely to be a faster and more cost-effective method to target this pathway in human disease. Pyrvinium is an FDA-approved anthelmintic drug used to treat pinworms, which also suppresses Wnt/β-catenin signaling by activating the β-catenin destruction complex protein, CK1α. Here, we demonstrate that breast cancer cells with increased expression of the oncogene INPP4B, a PI3K regulator that promotes Wnt/β-catenin activation, are selectively sensitive to pyrvinium treatment in 2D and 3D culture. Therefore, Wnt inhibition using pyrvinium may be an effective strategy for treating human breast cancers with high INPP4B expression.

**Abstract:**

The majority of breast cancers are estrogen receptor-positive (ER^+^), and endocrine therapies that suppress ER signaling are the standard-of-care treatment for this subset. However, up to half of all ER^+^ cancers eventually relapse, highlighting a need for improved clinical therapies. The phosphoinositide phosphatase, INPP4B, is overexpressed in almost half of all ER^+^ breast cancers, and promotes Wnt/β-catenin signaling, cell proliferation and tumor growth. Here, using cell viability assays, we report that INPP4B overexpression does not affect the sensitivity of ER^+^ breast cancer cells to standard-of-care treatments including the anti-estrogen 4-hydroxytamoxifen (4-OHT) or the PI3Kα inhibitor alpelisib. Examination of four small molecule Wnt inhibitors revealed that ER^+^ breast cancer cells with INPP4B overexpression were more sensitive to the FDA-approved drug pyrvinium and a 4-OHT-pyrvinium combination treatment. Using 3D culture models, we demonstrated that pyrvinium selectively reduced the size of INPP4B-overexpressing ER^+^ breast cancer spheroids in the presence and absence of 4-OHT. These findings suggest that repurposing pyrvinium as a Wnt inhibitor may be an effective therapeutic strategy for human ER^+^ breast cancers with high INPP4B levels.

## 1. Introduction

Almost 80% of all breast cancers are estrogen receptor-positive (ER^+^) [1]. Endocrine therapies such as selective estrogen receptor modulators (SERMs), selective estrogen receptor downregulators (SERDs) or aromatase inhibitors are the standard-of-care treatment for this breast cancer subset, and these therapies have significantly reduced mortality and disease recurrence [2]. However, despite initial high sensitivity to endocrine therapies, up to 50% of ER^+^ breast cancer patients eventually relapse, and many develop advanced disease that is refractory to subsequent endocrine therapy [3]. This has led to the clinical development of endocrine-based combination therapies such as the SERD, fulvestrant, together with the cyclin dependent kinase 4/6 (CDK4/6) inhibitor palbociclib/PD-0332991, which collectively prolong survival of patients with advanced ER^+^ breast cancer [4]. However, these treatments are not curative and affected individuals eventually succumb to disease, highlighting a clinical need to further improve therapeutics that target ER^+^ breast cancer.

The most common genetic event in ER^+^ breast cancer is mutation of *PIK3CA*, the gene encoding the p110α catalytic subunit of phosphoinositide 3-kinase alpha (PI3Kα), which occurs in up to 40% of ER^+^ breast cancers [5,6]. PI3Kα is activated by growth factor stimulation to generate the lipid second messenger, PI(3,4,5)P_3_, which is hydrolyzed to PI(3,4)P_2_ by inositol polyphosphate 5-phosphatases. PI(3,4,5)P_3_ and PI(3,4)P_2_ activate a number of downstream effectors including the serine/threonine kinase AKT to regulate cell growth, proliferation, survival and migration [7]. The most common *PIK3CA* mutations, *PIK3CA*^H1047R^ and *PIK3CA*^E545K^, are gain-of-function mutations that promote constitutive PI3Kα activation in the absence of growth factor stimulation and cause de novo murine mammary tumor formation [8,9,10]. For decades, there has been significant interest in the development of therapeutics that target PI3Kα signaling in breast cancer. The PI3Kα inhibitor alpelisib/BYL719 was recently approved to treat patients with advanced *PIK3CA*-mutant, hormone receptor-positive breast cancer in combination with fulvestrant [11]. The PI3Kα/γ/δ inhibitor, taselisib, in combination with fulvestrant, exhibits clinical benefit for metastatic ER^+^ breast cancers associated with multiple *PIK3CA* mutations [12]. However, current PI3K therapies cause significant adverse effects, and breast cancers can develop therapeutic resistance through a variety of mechanisms including acquired PTEN loss or insulin feedback [13,14].

Inositol polyphosphate 4-phosphatase type II (INPP4B) is a dual specificity 4-phosphatase that hydrolyzes PI(3,4)P_2_ to PI(3)P downstream of PI3Kα signaling [15]. INPP4B was initially identified as a tumor suppressor in ER^−^ basal-like breast cancer that suppresses PI(3,4)P_2_-mediated AKT signaling [16,17,18]. However, INPP4B paradoxically functions as an oncogene in several cancers, including acute myeloid leukemia (AML) and colon cancer where its overexpression promotes chemoresistance and tumor growth [19,20,21,22]. More recently, INPP4B was identified as a bona fide marker of ER-positivity in human breast cancer [23]. We reported that INPP4B expression is increased in up to 46% of ER^+^ breast cancer relative to normal breast tissue, correlating with mutant *PIK3CA* expression [24]. INPP4B overexpression promotes *PIK3CA*-mutant ER^+^ breast cancer cell proliferation and tumor growth via activation of Wnt/β-catenin signaling [24], suggesting that INPP4B functions as an oncogene in this context. There are currently no described INPP4B inhibitors available, and given its capacity to function as both a tumor suppressor and oncogene, therapeutically targeting this lipid phosphatase presents a significant clinical challenge. However, cell culture treatment with small molecule Wnt inhibitors that target PORCN or TNK rescued the increased proliferation of INPP4B-overexpressing ER^+^ breast cancer cells [24], which raises the possibility that ER^+^ breast cancers with high INPP4B expression may respond to therapies that target the Wnt/β-catenin pathway.

Here, we examined whether increased INPP4B expression alters the sensitivity of ER^+^ breast cancer cells to current standard-of-care therapies, and investigated the efficacy of Wnt therapeutics. INPP4B overexpression had little effect on the sensitivity of ER^+^ breast cancer cells to the SERM afimoxifene/4-hydroxytamoxifen (4-OHT) or the PI3Kα inhibitor alpelisib when cells were grown in monolayer cultures. However, INPP4B overexpressing ER^+^ breast cancer cells were more sensitive than control cells to pyrvinium, an FDA-approved drug that suppresses Wnt/β-catenin signaling, and to 4-OHT-pyrvinium combination treatment. In three-dimensional (3D) culture models, pyrvinium selectively reversed the increased size of INPP4B-overexpressing ER^+^ breast cancer spheroids in the presence and absence of 4-OHT. Collectively, our findings suggest that repurposing pyrvinium to inhibit Wnt/β-catenin signaling may be an effective strategy to selectively target ER^+^ breast cancers with high INPP4B expression.

## 2. Materials and Methods

### 2.1. Cell Culture

MCF-7 (cat # HTB-22) and T47D (cat # HTB-133) cells were purchased from ATCC. MCF-7 and T47D cells with stable expression of GFP-INPP4B or GFP-vector were generated and validated previously [24,25]. MCF-7 cells were grown in DMEM containing 10% (*v*/*v*) FCS, 2 mM L-glutamine, 100 units/mL penicillin, 1% (*v*/*v*) streptomycin, and 10 µg/mL insulin. T47D cells were grown in RPMI containing 10% (*v*/*v*) FCS, 2 mM L- glutamine, 100 units/mL penicillin, 1% (*v*/*v*) streptomycin, and 10 μg/mL insulin. All cells were maintained in a 5% CO_2_-humidified 37 °C incubator. All aseptic culture techniques were performed in a class II biohazard hood. Cells were routinely tested to confirm the absence of mycoplasma contamination. Cell line authentication was not performed.

For drug treatments, 2.4 × 10^4^ MCF-7 cells or 1.2 × 10^4^ T47D cells were seeded per well in a 48-well plate. The following day, cells were treated with 4-OHT (Selleckchem, cat # S7827), alpelisib (Selleckchem, cat # S2814), LGK-974 (Selleckchem, cat # S7143), PRI-724 (Selleckchem, cat # S8968), pyrvinium (Selleckchem, cat # S5816), celecoxib (Selleckchem, cat # S1261) or the same volume of DMSO as a vehicle control diluted in reduced growth media containing DMEM with 5% (*v*/*v*) FCS, 2 mM L-glutamine, 100 units/mL penicillin, 1% (*v*/*v*) streptomycin and 5 µg/mL insulin for MCF-7 cells, or RPMI containing 5% (*v*/*v*) FCS, 2 mM L- glutamine, 100 units/mL penicillin, 1% (*v*/*v*) streptomycin, and 5 μg/mL insulin for T47D cells.

### 2.2. Cell Viability Assays

Cell viability was assessed using the CellTiter-Glo^®^ 3D Cell Viability Assay (Promega, cat # G9683) according to the manufacturer’s protocol. Briefly, drug treatment media was removed and 100 μL of fresh reduced growth media then 100 μL CellTiter reagent were added to each well. The plate was shaken for 5 min at 500 rpm, then incubated for 30 minutes in the dark. A BMG LABTECH PHERAstar Plus plate reader with PHERAstar version 5.41 and MARS version 3.32 software was used to measure the integrated luminescence signal.

### 2.3. Hoechst/Propidium Iodide Staining

Drug treatment media was removed and replaced with phenol red-free media containing 1 µg/mL propidium iodide (PI) (Sigma, cat # P4170) and 1 µg/mL Hoechst 33342 (Bio-Rad, cat # 639) for 30 min. Cells were imaged using an Invitrogen EVOS^®^ FL Auto Imaging System with a 10× objective. The percentage of PI-positive cells was determined using the “cell counter” plugin in ImageJ version 2.0.0 software [26].

### 2.4. Ki67 Immunofluorescence

Approximately 2.4 × 10^4^ MCF-7 cells or 1.2 × 10^4^ T47D cells were seeded per well in a 48-well plate. The following day, cells were treated with reduced growth media containing 4-OHT, pyrvinium or DMSO. After 48 h of treatment, cells were fixed in 4% (*w*/*v*) paraformaldehyde (PFA). Cells were washed three times in PBS, then permeabilized for 15 min in PBS containing 0.1% (*v*/*v*) Triton X-100. Cells were washed three times in PBS, then were blocked for 1 h in PBS containing 3% (*w*/*v*) BSA. Cells were incubated overnight at 4 °C with Ki67 antibodies (ThermoFisher Scientific, cat # RM-9106-S1, 1:200) diluted in PBS containing 3% (*w*/*v*) BSA. Cells were washed three times in PBS, then incubated for 1 h with donkey anti-rabbit Alexa Fluor^®^ 555 antibodies (Invitrogen, cat # A-31572, 1:500) and DAPI (Sigma, cat # D9542, 1:1000) diluted in PBS containing 3% (*w*/*v*) BSA. Cells were washed three times in PBS, then imaged using an Invitrogen EVOS^®^ M5000 Imaging System with a 10× objective. The percentage of Ki67-positive cells was determined using the “cell counter” plugin in ImageJ version 2.0.0 software.

### 2.5. RNA Analysis

RNA was extraction was performed using the Isolate II RNA extraction kit (Bioline, cat # BIO-52073) according to the manufacturer’s protocol. RNA concentration was determined using a NanoPhotometer^®^ NP80 spectrophotometer (Implen). cDNA was synthezised from approximately 200 ng of RNA using the iScript gDNA clear cDNA synthesis kit (Bio-Rad, cat # 172-5035) according to the manufacturers’ instructions. qRT-PCR was performed with the QuantiTect SYBR Green PCR Kit (Qiagen, cat # 204143) using a CFX384 Real Time PCR System (Bio-Rad) with DKK1 (Qiagen, Geneglobe ID # QT00009093), LEF1 (Qiagen, Geneglobe ID # QT00021133) or RRN18S (Qiagen, Geneglobe ID # QT00199367) primers, and analyzed using CFX Manager version 3.1 software (Bio-Rad). The relative expression of *DKK1* or *LEF1* were normalized to the loading control gene *RRN18S*, and quantified using the ΔΔCT method.

### 2.6. 3D Spheroid Assays

MCF-7 or T47D cells were seeded in a 48-well plate by resuspending 5000 cells in 25 µL of Matrigel per well, then 400 µL growth media was added to each well. Cells were cultured in growth media for 4 days to allow 3D spheroid formation, then treated with 4-OHT, pyrvinium or the same volume of DMSO as a vehicle control diluted in reduced growth media. After 72 h, 3D spheroids were imaged using an Invitrogen EVOS^®^ FL Auto Imaging System with a 4× objective. The area of the largest 32 spheroids per well was measured using ImageJ version 2.0.0 software.

### 2.7. Statistics and Reproducibility

Cells were randomly assigned to treatment or control groups. Sample size calculations or blinding were not performed. All experiments were performed three times. Statistical analysis was performed by two-tailed unpaired *t* test or one-way ANOVA with Tukey post hoc test using Prism software version 7.0 (GraphPad). *p* values < 0.05 were considered statistically significant.

## 3. Results

### 3.1. INPP4B Has Minimal Effect on ER^+^ Breast Cancer Cell Sensitivity to 4-OHT or Alpelisib in Monolayer Culture

INPP4B enhances the chemoresistance of some cancers including acute myeloid leukemia and colorectal cancer [19,20,21]. As INPP4B expression is increased in *PIK3CA*-mutant ER^+^ breast cancers where it promotes cell proliferation and tumor growth [24], we questioned whether INPP4B also affects the sensitivity of ER^+^ breast cancer cells to standard-of-care therapies. We previously generated and validated MCF-7 and T47D human immortalized *PIK3CA*-mutant ER^+^ breast cancer cell lines with stable ectopic GFP-INPP4B overexpression, in order to model the effects of increased INPP4B expression as we described in a subset of ER^+^ breast tumors [24,25]. Here, we assessed the viability of cells grown in 2D monolayer culture in response to 48 h of treatment with the SERM 4-OHT, an active metabolite of tamoxifen that inhibits estrogen signaling and is a standard-of-care treatment for ER^+^ breast cancers. 4-OHT treatment reduced GFP-INPP4B and GFP-vector MCF-7 cell viability to a similar extent at concentrations of 10 µM or greater (Figure 1a,b). 4-OHT treatment also impeded GFP-INPP4B and GFP-vector T47D cell viability comparably (Figure 1c,d), suggesting that INPP4B overexpression does not affect ER^+^ breast cancer cell sensitivity to anti-estrogen therapy.

Combination therapy comprising the SERM fulvestrant and the PI3Kα inhibitor alpelisib was recently approved for advanced ER^+^ breast cancers that harbor *PIK3CA* mutations [11]. Thus, we also assessed whether INPP4B overexpression confers sensitivity of *PIK3CA*-mutant ER^+^ breast cancer cells to alpelisib. However, GFP-INPP4B expression did not affect MCF-7 or T47D cell viability to alpelisib treatment (Figure 1e–h). Cell viability was also assessed in response to 4-OHT-alpelisib combination treatment to mimic the effects of fulvestrant-alpelisib therapy used clinically. Interestingly, GFP-INPP4B expression modestly reduced MCF-7 cell sensitivity to 4-OHT-alpelisib combination treatment compared to controls (Figure 1i,j). However, no significant difference was observed between GFP-INPP4B and GFP-vector expressing T47D cells under comparable conditions (Figure 1k,l). Altogether, these data indicate that INPP4B has minimal effect on the sensitivity of *PIK3CA*-mutant ER^+^ breast cancer cells in monolayer culture to estrogen or PI3Kα-targeted therapies.

### 3.2. INPP4B Increases ER^+^ Breast Cancer Cell Sensitivity to Pyrvinium in Monolayer Culture

To determine whether INPP4B overexpressing ER^+^ breast cancer cells exhibit enhanced sensitivity to Wnt-targeted therapies, four small molecule Wnt inhibitors that target different Wnt pathway components and are clinically approved and/or undergoing clinical trials were evaluated. LGK-974 is a PORCN inhibitor that prevents Wnt ligand acylation and secretion and successfully reduced Wnt signaling in phase I clinical trials in a range of solid tumors including breast cancer [27,28]. PRI-724 disrupts the interaction between β-catenin and CBP and has also completed phase I trials for hepatitis virus-related cirrhosis [29,30]. Celecoxib/SC-58635 is an FDA-approved non-steroidal anti-inflammatory drug (NSAID) that inhibits COX-2, which like several other NSAIDs has also been reported to suppress Wnt/β-catenin signaling [31,32]. Pyrvinium is an FDA-approved anthelmintic drug used to treat pinworms that binds and hyperactivates the β-catenin destruction complex protein CK1α [33]. Here, we examined the effect of these candidate Wnt inhibitors on MCF-7 ER^+^ breast cancer cell viability after 48 h treatment in monolayer culture, and whether INPP4B expressing cells were more sensitive to these drugs. LGK-974 treatment had little effect on GFP-INPP4B or GFP-vector cell viability even at concentrations of up to 20 µM (Figure 2a,b), whereas PRI-724 treatment reduced the viability of GFP-INPP4B and GFP-vector cells in the low micromolar range (Figure 2c,d). Celecoxib had little effect on GFP-INPP4B and GFP-vector cell viability except at high doses of 50 µM (Figure 2e,f). No difference in viability was observed between GFP-INPP4B and GFP-vector MCF-7 cells following LGK-974, PRI-724 or celecoxib treatment (Figure 2a–f). Interestingly, pyrvinium reduced the viability of GFP-INPP4B and GFP-vector cells at nanomolar concentrations, and INPP4B expression significantly increased sensitivity to pyrvinium (Figure 2g,h). Thus, our analysis reveals that INPP4B overexpressing MCF-7 cells exhibit increased sensitivity to pyrvinium, but not to LGK-974, PRI-724 or celecoxib.

As therapeutics for ER^+^ breast cancer can be used clinically in combination with endocrine therapy [4,11], we investigated the effect of candidate Wnt inhibitors on cell viability in combination with 4-OHT. Interestingly, 4-OHT-LGK-974 reduced the viability of GFP-INPP4B cells but had little effect on GFP-vector controls (Figure 3a,b). 4-OHT-PRI-724 treatment reduced the viability of both GFP-INPP4B and GFP-vector cells, and GFP-INPP4B cells were significantly more sensitive to this treatment although the effects were modest (Figure 3c,d). GFP-INPP4B and GFP-vector cells exhibited similar sensitivity to 4-OHT-celecoxib treatment (Figure 3e,f). 4-OHT-pyrvinium treatment reduced GFP-INPP4B and GFP-vector cell viability even at very low nanomolar doses of pyrvinium, and importantly, GFP-INPP4B cells were markedly more sensitive (Figure 3g,h). Altogether, these findings suggest that pyrvinium may be a potential drug candidate for ER^+^ breast cancer cells with high INPP4B expression.

To determine whether pyrvinium treatment affects ER^+^ breast cancer cell death and/or proliferation, MCF-7 and T47D cells in monolayer culture were co-stained Hoechst/propidium iodide (PI) to detect dying cells (Figure 4), or Ki67 to detect proliferating cells (Appendix A). In the absence of treatment, GFP-INPP4B overexpression had no effect on MCF-7 or T47D cell death, but enhanced T47D cell proliferation. Treatment with 50 nM pyrvinium alone did not induce GFP-INPP4B or GFP-vector expressing MCF-7/T47D cell death. However, GFP-INPP4B expressing MCF-7 cell proliferation was reduced compared to GFP-vector controls under these conditions. Treatment with 10 µM 4-OHT induced cell death and suppressed proliferation of GFP-INPP4B or GFP-vector MCF-7/T47D cells in a similar manner. Combined 4-OHT-pyrvinium treatment had no further effect on MCF-7/T47D cell proliferation, but significantly increased the GFP-INPP4B cell death compared to GFP-vector controls. In control studies, GFP-INPP4B expressing MCF-7 and T47D cells exhibited increased expression of the Wnt target genes *DKK1* and *LEF1*, as previously reported [15], which was significantly reduced after 24 h treatment with 50 nM pyrvinium, confirming inhibition of Wnt signaling (Appendix A). Together, these data identify pyrvinium as a promising Wnt inhibitor for targeting ER^+^ breast cancers with high INPP4B expression, that is effective at low nanomolar concentrations in combination with 4-OHT.

### 3.3. Pyrvinium Selectively Reduces the 3D Growth of ER^+^ Breast Cancer Spheroids with INPP4B Overexpression

Breast cancers grow as 3D tumors where cellular morphology, spatial arrangement of cancer cells and their interaction with the tumor microenvironment impacts on how tumors respond to certain therapies. Although monolayer cancer cell cultures are a valuable experimental model for the rapid screening and evaluation of drug combinations, 3D cancer cell cultures are a superior pre-clinical model of tumor behavior in response to anti-cancer agents [34]. In order to assess whether INPP4B overexpressing ER^+^ breast cancer cells are sensitive to pyrvinium in 3D culture models, MCF-7 or T47D cells expressing GFP-vector or GFP-INPP4B were cultured in Matrigel for 4 days to form 3D spheroids, then treated with pyrvinium and/or 4-OHT for 72 h (Figure 5). As expected, GFP-INPP4B expressing MCF-7 and T47D cells formed larger spheroids than GFP-vector controls in the absence of pyrvinium or 4-OHT, consistent with increased cell proliferation as previously reported [24]. 4-OHT treatment decreased the size of GFP-vector spheroids but this effect was reduced for GFP-INPP4B spheroids which were significantly larger than 4-OHT-treated GFP-vector spheroids. In contrast, pyrvinium treatment had little effect on GFP-vector spheroids but rescued the increased GFP-INPP4B spheroid size. 4-OHT-pyrvninium treatment had no additive effect over 4-OHT treatment in GFP-vector spheroids, but significantly reduced the size of GFP-INPP4B spheroids to a level similar to 4-OHT-pyrvninium-treated GFP-vector spheroids. Collectively, these findings reveal that ER^+^ breast cancer cell lines with INPP4B overexpression exhibit increased sensitivity to the FDA-approved Wnt inhibitor pyrvinium or 4-OHT-pyrvinium combination treatment, and thus repurposing pyrvinium may be a therapeutic strategy to consider for this subset of breast cancers.

## 4. Discussion

INPP4B expression is increased in almost half of all ER^+^ breast cancers, which promotes cell proliferation and tumor growth via activation of Wnt/β-catenin signaling [24]. There are currently no therapeutics that directly target INPP4B. Here, we identify the FDA-approved Wnt inhibitor pyrvinium as a potential adjunct treatment for ER^+^ breast cancers with increased INPP4B expression. INPP4B overexpression did not affect the sensitivity of ER^+^ breast cancer cells in monolayer culture to current standard-of-care therapies including 4-OHT or alpelisib. However, INPP4B-overexpressing ER^+^ breast cancers cells were more sensitive to pyrvinium treatment when grown in monolayer culture, and combined 4-OHT-pyrvinium treatment significantly enhanced INPP4B-overexpressing ER^+^ breast cancer cell death. Pyrvinium or 4-OHT-pyrvinium treatment also rescued the increased size of INPP4B-overexpressing ER^+^ breast cancer 3D spheroids. Therefore, pyrvinium has the potential to be repurposed as a therapeutic for ER^+^ breast cancers that exhibit increased INPP4B expression.

Wnt/β-catenin signaling is hyperactivated in a range of human cancers including in up to 50% of breast cancers [35]. There has been substantial interest in developing therapeutics that suppress Wnt/β-catenin activation including small molecule inhibitors such as LGK-974 and PRI-724, as well as SFRP peptides and FZD antibodies [36]. However, the clinical progress of Wnt-targeted therapies has been impaired due to intolerable adverse effects, an inability to achieve sufficient Wnt inhibition in patients, and a lack of biomarkers to stratify patients that are more likely to respond to these therapies. Indeed, our results indicate that INPP4B-overexpressing ER^+^ breast cancers cells exhibit increased sensitivity towards LGK-974 or PRI-724 in combination with 4-OHT, suggesting that combining Wnt therapeutics with an endocrine therapy may be a potential strategy for this breast cancer subset. However, the effects observed in these experiments were modest, and required micromolar concentrations of LGK-974 or PRI-724 that are potentially difficult to achieve in affected individuals, suggesting that the specific Wnt inhibitors examined in this study may not be suitable for clinical translation in this context.

A promising clinical strategy for developing Wnt therapeutics is the repurposing of existing FDA-approved drugs that suppress Wnt signaling, thereby significantly reducing the time and cost of drug development. COX-2 inhibitors are widely used to treat inflammation and provide pain relief, and can suppress Wnt/β-catenin signaling via regulation of prostaglandin E2 (PGE2) [31,37]. Interestingly, our results indicate that MCF-7 cells with or without INPP4B overexpression exhibit little sensitivity towards the COX-2 inhibitor, celecoxib, even at micromolar concentrations. Although studies report that MCF-7 cells are sensitive to COX-2 inhibitors [38,39], others suggest that COX-2 inhibitors have little effect due to negligible COX-2 expression [40], consistent with our observations.

Pyrvinium is an anthelmintic drug that was FDA-approved in the 1950s to treat pinworm infections. More recently, pyrvinium was shown to bind and activate CK1α, a member of the β-catenin destruction complex, and thereby suppresses Wnt/β-catenin signaling [33], Consistent with this, our findings demonstrate that pyrvinium or 4-OHT-pyrvinium treatment selectively reduce the viability of ER^+^ breast cancer cells with INPP4B overexpression. Importantly, pyrvinium was effective at nanomolar doses and much lower concentrations than LGK-974 or PRI-724. As LGK-974, PRI-724 and pyrvinium all target different components of the Wnt pathway, our data would suggest that INPP4B-overexpressing cells may be more sensitive to CK1α inhibition by pyrvinium compared to PORCN or β-catenin/CBP inhibition by LGK-974 or PRI-724, respectively. As INPP4B promotes ER^+^ breast cancer cell proliferation and tumor growth via enhanced Wnt/β-catenin signaling [24], we predict that pyrvinium reduces cell viability via Wnt/β-catenin suppression. Indeed, pyrvinium treatment reduced Wnt target gene expression in INPP4B-overexpressing cells indicating inhibition of Wnt/β-catenin signaling. However, as pyrvinium is also reported to disrupt mitochondrial electron transport chain complexes and Hedgehog signaling [41,42,43], we cannot exclude the possibility that inhibition of other pathways may also contribute to the cytotoxicity of pyrvinium in ER^+^ breast cancer cells. Currently there are no clinical trials for pyrvinium in human breast cancer, but it is noteworthy that pyrvinium exhibited cytotoxic effects towards pancreatic cancer cells and xenograft tumors by inhibiting mitochondria function [44], and is currently undergoing phase I clinical trials in human pancreatic cancers (ClinicalTrials.gov Identifier: NCT05055323). Furthermore, pyrvinium treatment was cytotoxic at nanomolar concentrations to ER^−^, inflammatory breast cancer cells and stem-like breast cancer cells by inhibiting Wnt/β-catenin and lipid anabolism [45,46,47]. Altogether, these findings suggest that pyrvinium may have therapeutic potential for multiple breast cancer subsets, and should be further evaluated in vivo in combination with current standard-of-care therapies in future studies.

## 5. Conclusions

INPP4B is a bona fide marker of ER-positivity and exhibits increased expression in up to 46% of ER^+^ breast cancers where it promotes Wnt-mediated cell proliferation and tumor growth. Here, we show INPP4B overexpression does not affect the sensitivity of ER^+^ breast cancer cells to current standard-of-care therapies, but results in enhanced sensitivity to the FDA-approved Wnt inhibitor pyrvinium or 4-OHT-pyrvinium combination treatment in monolayer and 3D cultures. These findings raise the possibility that INPP4B could be used as a biomarker to stratify the subset of ER^+^ breast cancer patients that are likely to respond to adjunct pyrvinium therapy.

## Figures and Tables

**Figure 1 cancers-15-00135-f001:**
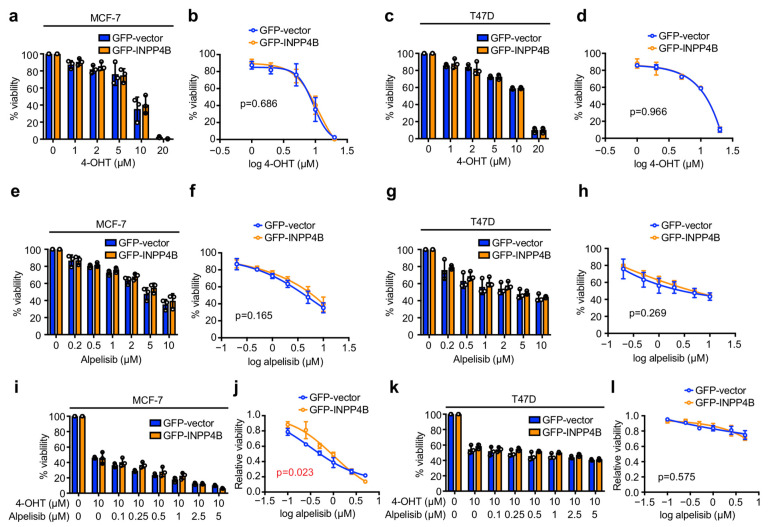
INPP4B overexpression has minimal effect on ER^+^ breast cancer sensitivity to 4-OHT or alpelisib in monolayer culture. (**a**–**d**) MCF-7 (**a**,**b**) or T47D (**c**,**d**) cells expressing GFP-vector or GFP-INPP4B were treated with 1–20 µM 4-OHT or DMSO as a vehicle control for 48 h, then cell viability was assessed using CellTiter-Glo^®^ assays. Data represent the relative cell viability normalized to untreated cells (**a**,**c**), or plotted as a 4-parameter logistical regression (**b**,**d**) ± SD (*n* = 3 experiments). (**e**–**h**) MCF-7 (**e**,**f**) or T47D (**g**,**h**) cells expressing GFP-vector or GFP-INPP4B were treated with 0.2–10 µM alpelisib or DMSO as a vehicle control for 48 h, then cell viability was assessed using CellTiter-Glo^®^ assays. Data represent the relative cell viability normalized to untreated cells (**e**,**g**), or plotted as a 4-parameter logistical regression (**f**,**h**) ± SD (*n* = 3 experiments). (**i**–**l**) MCF-7 (**i**,**j**) or T47D (**k**,**l**) cells expressing GFP-vector or GFP-INPP4B were treated with 10 µM 4-OHT and 0.1–5 µM alpelisib or DMSO as a vehicle control for 48 h, then cell viability was assessed using CellTiter-Glo^®^ assays. Data represent the relative cell viability normalized to untreated cells (**i**,**k**), or plotted as a 4-parameter logistical regression normalized to 4-OHT-treated cells (**j**,**l**) ± SD (*n* = 3 experiments). *p* values were determined by two-tailed unpaired *t* test of the area under the curve in (**b**,**d**,**f**,**h**,**j**,**l**).

**Figure 2 cancers-15-00135-f002:**
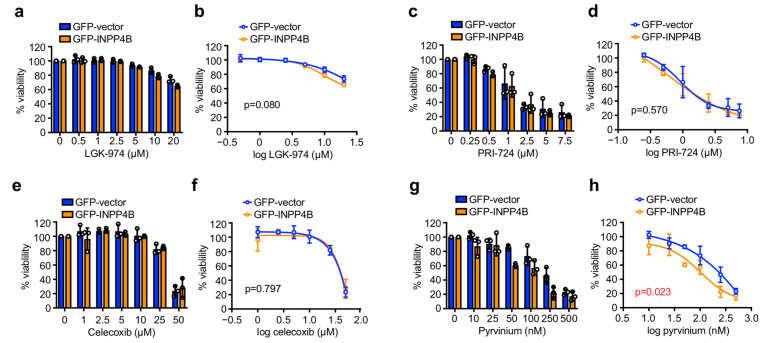
INPP4B-overexpressing MCF-7 cells are more sensitive to pyrvinium in monolayer culture. MCF-7 cells expressing GFP-vector or GFP-INPP4B were treated with 0.5–20 µM LGK-974 (**a**,**b**), 0.25–7.5 µM PRI-724 (**c**,**d**), 1–50 µM celecoxib (**e**,**f**), 10–500 nM pyrvinium (**g**,**h**) or DMSO as a vehicle control for 48 h, then cell viability was assessed using CellTiter-Glo^®^ assays. Data represent the relative cell viability normalized to untreated cells (**a**,**c**,**e**,**g**), or plotted as a 4-parameter logistical regression (**b**,**d**,**f**,**h**) ± SD (*n* = 3 experiments). *p* values were determined by two-tailed unpaired *t* test of the area under the curve in (**b**,**d**,**f**,**g**).

**Figure 3 cancers-15-00135-f003:**
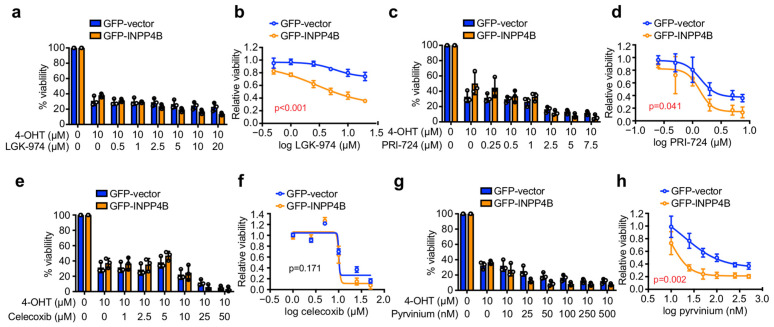
INPP4B-overexpressing MCF-7 cells are more sensitive to 4-OHT-pyrvinium combination treatment in monolayer culture. MCF-7 cells expressing GFP-vector or GFP-INPP4B were treated with 10 µM 4-OHT ± 0.5–20 µM LGK-974 (**a**,**b**), 0.25–7.5 µM PRI-724 (**c**,**d**), 1–50 µM celecoxib (**e**,**f**), 10–500 nM pyrvinium (**g**,**h**) or DMSO as a vehicle control for 48 h, then cell viability was assessed using CellTiter-Glo^®^ assays. Data represent the relative cell viability normalized to untreated cells (**a**,**c**,**e**,**g**), or plotted as a 4-parameter logistical regression normalized to 4-OHT-treated cells (**b**,**d**,**f**,**h**) ± SD (*n* = 3 experiments). *p* values were determined by two-tailed unpaired *t* test of the area under the curve in (**b**,**d**,**f**,**g**).

**Figure 4 cancers-15-00135-f004:**
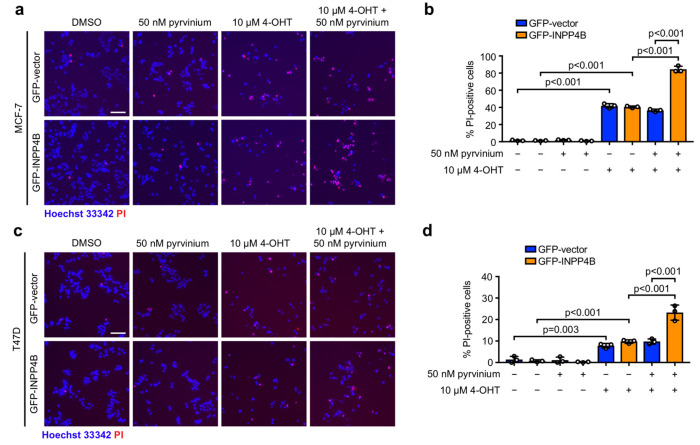
4-OHT-pyrvinium treatment selectively enhances INPP4B-overexpressing ER^+^ breast cancer cell death. (**a**–**d**) MCF-7 (**a**,**b**) or T47D (**c**,**d**) cells expressing GFP-vector or GFP-INPP4B were treated with 50 nM pyrvinium and/or 10 µM 4-OHT or DMSO as a vehicle control for 48 h, then incubated with 1 µg/mL propidium iodide (PI) and 1 µg/mL Hoechst 33,342 for 30 min (**a**,**c**). Data represent the percentage of PI-positive cells ± SD (*n* = 3 experiments, >300 cells/experiment) (**b**,**d**). Scale bar is 100 µm in **a**, **c**. + indicates where treatment was added, and – indicates where no treatment was added. *p* values were determined by one-way ANOVA with Tukey post hoc test in (**b**,**d**).

**Figure 5 cancers-15-00135-f005:**
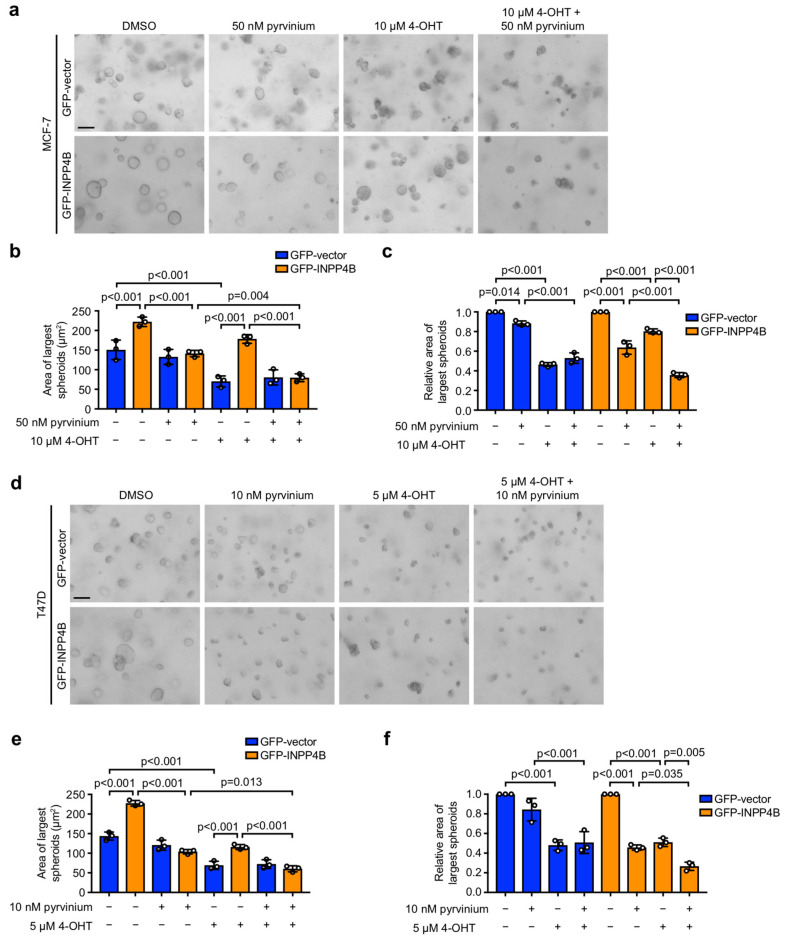
INPP4B-overexpressing ER^+^ breast cancer spheroids are more sensitive to pyrvinium or 4-OHT-pyrvinium. (**a**–**c**) MCF-7 cells expressing GFP-vector or GFP-INPP4B were cultured in Matrigel for 4 days to form 3D spheroids, then treated with 50 nM pyrvinium and/or 10 µM 4-OHT or DMSO as a vehicle control for 72 h. Representative brightfield images of 3D spheroids are shown (**a**). Data represent the area of the largest 32 spheroids (**b**), or the relative area of the largest 32 spheroids normalized to untreated cells (**c**) ± SD (*n* = 3 experiments). (**d**–**f**) T47D cells were cultured in Matrigel for 4 days to form 3D spheroids, then treated with 10 nM pyrvinium ± 5 µM 4-OHT or DMSO as a vehicle control for 72 h. Representative brightfield images of 3D spheroids are shown (**d**). Data represent the area of the largest 32 spheroids (**e**), or the relative area of the largest 32 spheroids normalized to untreated cells (**f**) ± SD (*n* = 3 experiments). Scale bar is 100 µm in (**a**,**d**). + indicates where treatment was added, and – indicates where no treatment was added. *p* values were determined by one-way ANOVA with Tukey post hoc test in (**b**,**c**,**e**,**f**).

## Data Availability

This study does not include data deposited in external repositories. All data that support the findings of this study are available from the corresponding author upon reasonable request.

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
