# Peer review of "The FDA-Approved Drug Pyrvinium Selectively Targets ER+ Breast Cancer Cells with High INPP4B Expression"

_cancers, 2022, doi:10.3390/cancers15010135_

Round 1

Reviewer 1 Report

Samuel J. Rodgers and colleagues describe in the present paper the impact of the  phosphoinositide phosphatase INPP4B overexpression  in estrogen positive breast cancer cell lines. Thus they show the INPP4B over expression is associated with an increase sensitivity to pyrivinium but no significant differences are observed with anti-estrogen therapy or with a PI3Ka inhibitor alpelisib. Moreover they also pointed out a reduction of INPP4B over expressing ER+ breast cancer spheroid size after a treatment with pyrivinium and 4-OHT.

- My major concern is based on the fact that the authors have only used a single viability test « Cell titer Glo » which is related to the metabolic level of cells. Thus, the used of other tests could improved their conclusions. In addition to other proliferation tests the authors have to include experiments demonstrating the increase of cell death not only in their 2D and 3D models. This is particularly important since pyrvinium acts through energy and autophagy depletion as well as inhibition of Akt and Wnt-β-catenin-dependent pathways.

- All experiments should include the second cell line T47D to validate the results obtained in MCF7 cells.

- Since the authors predict that pyrvinium dependent decrease of cell viability is due to Wnt/β-catenin suppression in the discussion part, experiments demonstrating this mode of action has to be performed.

Reviewer 2 Report

This Communication by Rodgers et al. reported new results showing that INPP4B overexpression in the ER+ cells sensitizes the cells to pyrvinium, an FDA-approved Wnt inhibitor treatment. Overall, this is a novel study with the proper amount and quality of data which may be sufficient for a communication paper. A few comments were listed below for consideration,

1.      Four small molecule Wnt inhibitors were screened but only pyrvinium has some syngeneic effect with INPP4B overexpression, why is that? A potential explanation or discussion should also be included.

2.      Some of the cell viability data is not consistent with the corresponding 4-parameter logistical regression curve. These include Figures 3, a and b, c and d. Please clarify.
